# Robocasting of Ceramic Fischer–Koch S Scaffolds for Bone Tissue Engineering

**DOI:** 10.3390/jfb14050251

**Published:** 2023-04-30

**Authors:** Vail Baumer, Erin Gunn, Valerie Riegle, Claire Bailey, Clayton Shonkwiler, David Prawel

**Affiliations:** 1Department of Mechanical Engineering, Colorado State University, Fort Collins, CO 80523, USA; 2Department of Computer Science, Colorado State University, Fort Collins, CO 80523, USA; 3School of Biomedical Engineering, Colorado State University, Fort Collins, CO 80523, USA; 4Department of Mathematics, Colorado State University, Fort Collins, CO 80523, USA

**Keywords:** 3D printing, ceramics, hydroxyapatite, tricalcium phosphate, TPMS, gyroid, Fischer-Koch, scaffold, robocasting, photocasting, tissue engineering

## Abstract

Triply Periodic Minimal Surfaces (TPMS) are promising structures for bone tissue engineering scaffolds due to their relatively high mechanical energy absorption, smoothly interconnected porous structure, scalable unit cell topology, and relatively high surface area per volume. Calcium phosphate-based materials, such as hydroxyapatite and tricalcium phosphate, are very popular scaffold biomaterials due to their biocompatibility, bioactivity, compositional similarities to bone mineral, non-immunogenicity, and tunable biodegradation. Their brittle nature can be partially mitigated by 3D printing them in TPMS topologies such as gyroids, which are widely studied for bone regeneration, as evidenced by their presence in popular 3D-printing slicers, modeling systems, and topology optimization tools. Although structural and flow simulations have predicted promising properties of other TPMS scaffolds, such as Fischer–Koch S (FKS), to the best of our knowledge, no one has explored these possibilities for bone regeneration in the laboratory. One reason for this is that fabrication of the FKS scaffolds, such as by 3D printing, is challenged by a lack of algorithms to model and slice this topology for use by low-cost biomaterial printers. This paper presents an open-source software algorithm that we developed to create 3D-printable FKS and gyroid scaffold cubes, with a framework that can accept any continuous differentiable implicit function. We also report on our successful 3D printing of hydroxyapatite FKS scaffolds using a low-cost method that combines robocasting with layer-wise photopolymerization. Dimensional accuracy, internal microstructure, and porosity characteristics are also presented, demonstrating promising potential for the 3D printing of TPMS ceramic scaffolds for bone regeneration.

## 1. Introduction

Bone fractures are a significant and growing healthcare issue in the United States. Poor healing of large bone defects is one of the biggest challenges in human orthopedic medicine, affecting more than 1.5 million Americans per year and often leading to infections, reoperations, poor functional outcomes, and ultimately, all too often, limb loss [1]. In 2015, this resulted in significant personal and economic cost of more than $1 billion per year [1]. By 2025, these figures are expected to rise by 50% [2]. Furthermore, additional indirect costs due to productivity loss are estimated at 42% of the direct expenditure on average. These indirect costs more than double for patients experiencing delayed or non-union healing [3]. Populations over 50 years of age experience these fractures more often than their younger counterparts, and fracture rates increase exponentially between ages 50 and 85 [4]. Extended life expectancies paired with bone diseases such as osteosarcoma and osteoporosis further contribute to this increased fracture rate and emphasize the importance of new therapeutic techniques.

Autologous bone grafting is the current gold standard procedure to remediate large bone defects, but reported outcomes are too often unsatisfactory. Healing strategies using bone grafts and bone fillers exhibit recurring failures with non-union rates as high as 21% [5], and complication rates of 50% due to delayed or non-union, 30% from allograft fracture, and 15% from infection [6]. Autografts are limited by availability and the size of harvestable tissue, and they create two surgical sites that are prone to injury, infection, and significant patient discomfort [7]. Synthetic scaffolds have emerged in recent decades as promising alternatives to bone grafting because they address some of these shortcomings.

Three-dimensional (3D) printing of scaffolds for bone tissue engineering (BTE) is a leading method to replace bone grafts [7] and is under significant investigation in pre-clinical settings [8]. Successful BTE scaffolds should be biocompatible, with surface characteristics that promote cell adhesion (osteoconductivity), proliferation (osteoinductivity) and mineralization (osteointegration). To meet these requirements, synthetic scaffolds have been produced in countless materials from metals such as titanium and ceramics such as calcium-phosphate-based materials to composites such as bioglass and polymeric blends [9]. Among these options, calcium phosphate-based materials, such as hydroxyapatite (HAp) and tricalcium phosphate (TCP), are popular for BTE due to their biocompatibility, high levels of bioactivity (osteoconductivity, osteoinductivity and osteointegration), compositional similarities to human bone mineral, non-immunogenicity, tunable degradation rates, and promising drug delivery capabilities [10,11,12,13,14]. 3D printing of ceramics has shown great potential, but the fabrication and design methodologies used in 3D printing remain limited in their ability to produce large-sized scaffolds for load bearing cases [15,16,17]. More research is needed into ceramic scaffold structure and fabrication.

To improve ceramic scaffolds, their complex requirements of porosity, interconnectivity, and strength to remediate large bone defects must be understood. BTE scaffolds attempt to replicate the properties of bone tissue surrounding a defect to encourage the body’s natural healing process. Scaffolds should have a high porosity to mimic natural trabecular bone, which has a honeycomb-like internal structure with a porosity range of 50–90%, depending on the anatomical location [18]. The porous interior of BTE scaffolds should form a continuous network to accelerate the mass transport of nutrients, gasses, and waste, thus augmenting the bone remodeling process [19]. Scaffolds lack the vasculature of autologous bone, making them solely dependent on diffusion for mass transport. This emphasizes the need for high levels of permeability [20], which is used as a functional representation of porosity, pore size, pore shape, tortuosity, and interconnectivity [21]. Lastly, scaffolds require structural integrity as regeneration processes develop new bone. However, the pore size, volume fraction, and porosity ranges required for ideal bioactivity are also associated with fairly poor mechanical properties [19,22,23,24]. Consequently, BTE scaffolds must balance mechanical properties (compression, stiffness and elasticity) against interconnectivity, porosity, and pore size to optimize for each application and the associated load bearing requirement [25,26]. Considerable progress has been made using ceramic and polymer/ceramic composite BTE scaffolds in critical defect healing, as demonstrated by recent in vivo studies [27,28,29,30]. But the challenge of complete bridging, integration and union remains for human-scale load-bearing cases.

Structural innovations such as triply periodic minimal surfaces (TPMS) are enabling the production of scaffolds that are stiffer and stronger than traditional rectilinear topologies [31,32,33,34,35]. TPMS are implicit functions with infinitely stackable 3D unit cells and relatively high porosity and strength per volume, which are proving to be ideal candidates for BTE due to their relatively high mechanical energy absorption and robustness, interconnected internal porous structure, scalable unit cell topology, and smooth internal surfaces with relatively high surface area per volume [31,32,36]. The TPMS function has zero mean curvature, which creates a continuous interior devoid of sharp corners and junctions within each unit cell. Additionally, the parameters can be adjusted to achieve specific porosities, pore sizes, shapes, permeability, and tortuosity that are favorable for BTE scaffolds [35,37,38,39,40,41,42,43]. A depiction of popular TPMS unit cells in BTE are shown in Figure 1 [39].

Advanced fabrication techniques such as additive manufacturing (a.k.a. 3D printing) are ideally suited to precisely and accurately reproduce the geometric and topological design constraints of TPMS scaffolds for BTE [9,14,44]. Numerous researchers have 3D printed gyroid and other TPMS structures using powder-bed fusion (PBF). Abueidda et al. used PBF to 3D print gyroid scaffolds in nylon to study their mechanical properties [32]. Abou-Ali et al. fabricated four different types of TPMS structures, including gyroid and Fischer–Koch S (FKS), in nylon using PBF in order to study TPMS mechanical properties [45]. In other applications of PBF, Maskery et al. manufactured gyroids composed of aluminum alloys [46] and other TPMS, including gyroids but not FKS, from nylon [41], also to study the mechanical properties of these scaffolds. Castro et al. printed three types of TPMS structures, including gyroid but not FKS, using vat photopolymerization to confirm their finite element simulation with mechanical behavior [47]. Melchels et al. 3D printed gyroid structures using vat photopolymerization to study the effects of scaffold architecture on cell proliferation [48]. Santos et al. studies the effects of permeability and porosity on four different TPMS structures (not including FKS) that were 3D printed using material jetting of a commercial photopolymeric material. [49] Many researchers have 3D printed gyroid and other TPMS (not FKS) structures by melt extruding poly-lactic acid (PLA), acrylonitrile butadiene styrene (ABS), and other materials [33,50,51]. The majority of research with FKS has been limited to computer simulation. Numerous researchers included FKS in their finite element analysis (FEA) and computational fluid dynamics (CFD) modeling of TPMS structures [35,39,52].

Work with ceramic materials using TPMS is comparatively limited. The Bose group used a binder jetting 3D-printing process to study pore size and pore volume effects on alumina and TCP gyroid scaffolds [23] and the mechanical and biological properties of HAp gyroid scaffolds [53]. In another study, Restrepo et al. used robocasting with an unspecified “conventional commercial ceramic paste” in their study of mechanical properties of three ceramic TPMS structures (not including FKS) [54]. The limited application of TPMS with ceramics results from challenges in production of these complex structures. Resources for creating printable gcode for FKS and other non-gyroid TPMS are non-existent, exclusive of proprietary software that is mostly bundled with expensive 3D printers and which generally use proprietary cytotoxic materials. Conversely, gyroids are core infill patterns in popular 3D-printing slicers such as Ultimaker Cura (Ultimaker B.V., Utrecht, Netherlands), and embedded topologies in CAD platforms such as Creo (Parametric Technologies Corp., Boston, MA, USA) and topology optimization tools such as nTopology (nTopology Inc., New York, NY, USA).

Robocasting is a promising method for fabricating ceramic BTE scaffolds because it is very low cost, easy to use, can create high-precision objects, and requires small amounts of ceramic material. The method uses evaporative processes to remove liquified polymeric or aqueous materials, within which, powdered solid content is carried to form the object. High solid loading required for mechanical strength is challenged by significant shrinkage and cracking that result from evaporation of these liquid carriers. Moreover, large overhangs are difficult to fabricate due to longer “drying” times as the solvent carriers dissipate. Such overhangs usually require sacrificial supports that can not be removed within the scaffold structure, which limits the ability of robocasting to form complex 3D objects with good structural strength, topological complexity, and model fidelity, which also contain completely bridged overhangs (e.g., interconnected pores). Complex structures such as TPMS scaffolds require the ability to print slurries without supports, and the viscous extruded material must harden quickly with limited shape distortion. From these challenging requirements a new approach has emerged that combines photopolymerization with robocasting in which layers are cured layer-by-layer as they are printed, eliminating the need for support material, and enabling fabrication of highly complex, high precision structures. We refer to this method as “photocasting.” Faes et al. was the first to use this approach to fabricate featureless slabs of yttrium-stabilized Zirconia [55]. Asif et al. [56] and Farahani et al. [57] produced basic structures consisting of a photopolymeric resin containing fumed silica particles. We used photocasting to print gyroids for BTE in earlier work [58,59].

It has been suggested that some TMPS topologies might be better suited to particular applications. Lu et al. presented computer simulations comparing the properties of different TPMS topologies, including FKS [39]. Their finite-element analysis suggested the FKS topology may be better suited to remediate a cortical diaphyseal bone defect due to its isotropic behavior, even at high porosities [60]. Because cortical bone can be modeled as an isotropic tissue in certain defects, the specific properties of FKS might be better suited to this application than other TPMS, such as gyroid scaffolds. Similarly, they proposed that gyroid scaffolds might be better suited to procedures such as epiphyseal tibial tuberosity advancement, where the higher anisotropic properties of the gyroid may be preferable to match the anisotropic mechanical behaviors of cancellous bone [60]. Lu concludes that the FKS topology “may be the most favorable one in the scenarios where nutrient is not limiting, e.g., in the application of bone fusion.” Yet, the characteristics of FKS ceramic scaffolds have yet to be explored in the context of BTE, probably due in large part to the production issues mentioned above. To the best of our knowledge, no one has 3D printed FKS scaffolds in calcium phosphate-based materials. Abou-Ali [45] appears to be the only researcher to have actually 3D printed FKS, but not for a BTE application and using an expensive, commercial photopolymer printer and a material that is unlikely to be biocompatible.

This paper addresses clear roadblocks to producing FKS and other TPMS scaffolds by developing and utilizing open-source software on low-cost 3D printers to produce ceramic scaffolds with features in the requirement range for BTE. The purpose of our work is to enable experimentation and innovation with FKS and other TPMSs by more researchers, not the optimization of ceramic FKS scaffolds for BTE. We report on a software algorithm that we developed to create 3D printable scaffold models for TPMS such as FKS and gyroids and any continuous differentiable implicit function representation, also including spheres, ellipsoids, and conic sections. We contributed the software to the open-source community so that others can model and print these complex topologies, along with mixed and functionally graded topologies. We then used this algorithm to 3D print FKS scaffolds for BTE using photocasting. In addition, we evaluated the accuracy of our fabrication process in the context of BTE for dimensional control, porosity, pore size, and surface texture.

## 2. Materials and Methods

### 2.1. Creating a Flexible, Customizable 3D TPMS Model for BTE

To generate the Fisher–Koch S triply periodic minimal surface, the trigonometric approximation in Equation (1) was applied. This approximation is an implicit function where the surface is defined to exist when $fks\left(x,y,z\right)=0$.
(1)$$fks\left(x,y,z\right)=\cos\left(2x\right)\sin\left(y\right)\cos\left(z\right)+\cos\left(2y\right)\sin\left(z\right)\cos\left(x\right)+\cos\left(2z\right)\sin\left(x\right)\cos\left(y\right)$$

A mathematical surface has zero thickness and must be converted to a solid volume for realization. A sheet solid methodology was used where the surface was continuously expanded in opposing normal directions to create a uniform thickness centered on the curvature. To tune scaffold properties for BTE, a method was needed to create FKS objects with a variable thickness (k) divided into (m) number of material deposition paths because the ratio of k/m could be set to the extruder nozzle width for optimal slicing, and therefore optimal build quality. Furthermore, the material deposition paths must remain parallel for cohesive layers to form. However, offsetting an implicit surface by adding a constant k is insufficient for the trigonometric TPMS approximation functions, as the method does not offset the surface by a uniform distance. This results in filament deposition paths that both diverge and converge, leading to cavities and overlapping filament. Additionally, the offset surface is not centered around the desired distance k, causing the average filament line spacing to be incorrect for the desired line width. See Figure 2 for an illustration of these issues.

Geier et al. proposed a method of surface offsetting with significantly reduced line spacing variance by adding the magnitude of the gradient vector of the surface to the original implicit function [61]. This method works because the gradient is a proxy for the normal of a surface and it can be shown that the gradient and normal vectors are coaxial. While this first order approximation, as the offset k grows, is still prone to the same errors that are seen in the simple offset method previously mentioned, we found that the errors were significantly smaller with the Geier method, as visualized in Figure 2. For FKS, we added the gradient of Equation (2) and multiplied it by k, as shown in Equation (3).
(2)$$\nabla fks\left(x,y,z\right)=\left(\frac{\partial fks\left(x,y,z\right)}{\partial x},\frac{\partial fks\left(x,y,z\right)}{\partial y},\frac{\partial fks\left(x,y,z\right)}{\partial z}\right)$$
(3)$$fks_{offset}\left(x,y,z,k\right)=fks\left(x,y,z\right)-k\left|\nabla fks\left(x,y,z\right)\right|$$

Using the FKS approximation and offset function, the surface was generated and discretized into a non-manifold triangulated mesh by a commonly used marching cubes iso-contouring algorithm. The algorithm works by first generating a level set of the function, finding the points that intersect the zero level at neighboring cells, and finally, triangulating the points of a cell using a lookup table. In order to modify the mesh for BTE scaffolds, an open-source python GUI application [62] that used the Vedo python library [63] was built for customization, visualization, and mesh processing. This exportable mesh was designed to leverage the ‘Surface Mode’ of the Ultimaker Cura slicing software to repair gaps and slice the object into universal gcode instructions for viscous extrusion by our 3D printer(s).

The GUI is demonstrated in Figure 3, which shows that users can select a TPMS topology and then adjust the dimensions, periodicity, road width, and road count to a custom porosity calculated according to Equation (4). The designed porosity of a TPMS scaffold ($\phi_{d}$) can be estimated by dividing the estimated void volume, $V_{v}$, of the scaffold by the total volume of the scaffold, $V_{t}$. In our case, the total volume is the volume of the cube. We approximate the void volume by discretizing the domain into individual cells of volume, $V_{c}$, and then multiplying that volume by the number of cells, c, that exist outside the mesh, as shown in Equation (5). The cell size was decreased until the solution converged.
(4)$$\phi_{d}\;\%=\frac{V_{v}}{V_{t}}*100\%$$
(5)$$V_{v}=c\cdot V_{c}$$

The pore size in the context of BTE was defined as the diameter of the most constricted region on a 2D slice (in the XY plane), which was measured on the model of the exported mesh. For FKS specifically, there are multiple such regions where interconnected pores constrict to a circular cross section. Within the GUI, pore size can be adjusted by varying periodicity and wall thickness, where wall thickness is defined as the product of the road width and road count.

### 2.2. Scaffold Fabrication

#### 2.2.1. Scaffold Design

Using the FKS program described earlier, 1.4 periods of the FKS function were distributed over a 12 mm side length to result in cubic scaffolds with an as-designed porosity of 73.71%. This was selected to be in the middle of the approximate 50–90% range used in BTE. A single road width of 0.413 mm, equal to the nozzle diameter, was used in the design. These properties are herein referred to as the “as-designed” characteristics. In prior work [59] the sintering process showed an isotropic shrinkage of approximately 21% for HAp gyroid scaffolds. This shrinkage is an expected part of the process and will be analyzed for FKS scaffolds. The expected shrinkage influenced the choice for a designed 12 mm cube, thus targeting final as-sintered dimensions of roughly 10 mm × 10 mm × 10 mm.

#### 2.2.2. Slurry Preparation

For ceramic photocasting, a viscous mixture, referred to in this report as a slurry, was prepared to suspend needle-like HAp particles (Macron Fine Chemicals, Avantor, Radnor, PA, USA) in 99% pure ethylene glycol dimethacrylate (EGDMA, Scientific Polymer Products, Inc., Ontario, NY, USA) to enable controlled viscous extrusion. A photoinitiator, diphenyl(2,4,6-trimethylbenzoyl)phosphine oxide(TPO, TCI America, Portland, OR, USA) was added to permit later photocuring, and a commercial anionic dispersant, Solplus D540 (Lubrizol Advanced Materials Inc., Wickliffe, OH, USA), was added to reduce viscosity by dispersing the HAp particles in the monomer. The slurry composed of EGDMA, TPO, and D540 was then mixed with agate milling media in Teflon^®^ jars on a planetary ball mill (Across International, Davie, FL, USA) at 120–360 rpm, depending on process cycle, for several hours, while gradually adding additional HAp powder until a homogenous slurry with 41% volume HAp was achieved. The final slurry was then sealed in an airtight, opaque jar to avoid vaporization of EGDMA and premature photocuring until the 3D-printing process occurred.

#### 2.2.3. 3D Printing (“Photocasting”)

FKS scaffolds were photocast using a Hyrel Hydra (Hyrel 3D, Norcross, GA, USA) 3D printer. The slurry was loaded into a syringe print head assembly (HyRel EMO-XT print head) with a 22-gauge (0.413 mm) tip under limited light exposure, and then transferred to the printer. The print bed was covered with painter’s tape to improve bed adhesion and to prevent reflected light from prematurely curing slurry in the nozzle. The FKS scaffold g-code was uploaded to the printer and a ring LED surrounding the print nozzle was activated at 405nm wavelength to produce an exposure of 0.91 mW/cm^2^. The printer executed the machine code with continuous material deposition and layer-wise photopolymerization until the build was completed. Lastly, the scaffold was post-cured under LED for an additional 3 min before it was stored in darkness prior to subsequent sintering. The intermediate 3D-printed, but not yet sintered, scaffold is referred to as a “green body” and its dimensions are “as-printed.”

#### 2.2.4. Sintering

Green body scaffolds were sintered to remove organic content and to densify the HAp within the scaffold struts. Scaffolds were heated in a muffle furnace (Barnstead/Thermolyne 47900, Ramsey, MN, USA) at a ramp rate of 5 °C/min up to 1200 °C, and then held for 3 h. Thermogravimetric analysis confirmed in prior work that all polymeric content was removed [59]. The furnace cooled to ambient temperature, and the scaffolds were transferred to a cool, dry location to await characterization. These “as-sintered” scaffolds are the final product that were then used in further experimentation. 

### 2.3. Characterization of Manufactured Scaffold Structure

#### 2.3.1. Dimensional Accuracy

The “accuracy” of the 3D-printing process is defined as the fidelity of the 3D-printed model to the original 3D CAD model in the principal axes. Thirteen scaffold cubes were measured in three axes using calipers at the green body “as-printed” stage and at the final “as-sintered” stage to be compared against the as-designed (CAD) side length defined in the FKS software. The XY plane was denoted as the coordinate plane for each layer, and the Z axis described the vertical build direction.

#### 2.3.2. Porosity, Pore Size, and Wall Thickness

Micro-computed tomographic (micro-CT) imaging using a Scanco 80 (Scanco Medical AG, Bruttisellen, Switzerland) measured global scaffold porosity, pore size, and wall thickness of three sintered FKS scaffolds. In the Scanco 80 Visualizer, a pre-existing setting designed to scan porous, bone-like materials was used, and four key measurements were recorded from each scan: Total Volume (TV), Filled Volume (FV), Trabecular Thickness (Tb.Th), and Trabecular Spacing (Tb.Sp). The TV is manually selected as a region of interest and defines the nominal volume occupied by the scaffold under analysis. FV is the volume of ceramic component above a minimum density within the TV. TV and FV were used to determine sintered porosity ($\phi_{s}$) using Equation (5).
(6)$$\phi_{s}\;\%=\left(1-\frac{TV}{FV}\right)*100\%$$

Tb.Th is equivalent to the average wall thickness in 3D, and Tb.Sp describes the average spacing of those walls. Pore size was defined as the major and minor diameters of an ellipse that is compared with the most constricted pores in the XY plane. Measurements were taken using Image J [64] on 2D images from the scan data at six equally spaced slices in the Z direction per scaffold. The solid model generated from the scan data was further used to study the fidelity of the printed FKS topology and to view internal morphology.

#### 2.3.3. Scaffold Microstructure, Surface Morphology & Layer Cohesion

Scanning Electron Microscopy (SEM) on a JEOL JSM-6500F field emission scanning electron microscope (JEOL, Peabody, MA, USA) was utilized to observe the printed surface microstructure and morphology and to visualize the interfacial boundary region between roads. Scaffold samples were coated with 10nm of gold using a Denton Vacuum Desk II Gold Sputter Coater (Denton Vacuum, Moorestown, NJ, USA) and imaged at 10 to 15kV.

## 3. Results

### 3.1. Scaffold Fabrication

Figure 4 graphically diagrams the complete workflow through selecting parameters in the GUI, exporting the STL, slicing in Ultimaker Cura, and lastly, 3D printing. This workflow resulted in consistent fabrication of FKS scaffolds, an example of which is shown in Figure 4E, that accurately represented the as-designed scaffolds, as reported below.

Representative printed scaffolds are shown in Figure 5. Surfaces printed uniformly and overhangs bridged without notable collapse. Extruded roads were well-formed, as defined by their consistent width, unbroken length, and smooth surface texture. Road width varied most near the ends of roads in the vertical sides (X–Z and Y–Z planes) of the scaffold, where excess slurry extrusion, referred to as “ooze,” resulted in a melted appearance, as illustrated in the Z–Y plane of Figure 5A,B.

In Figure 6A, red paths represent the sliced road extrusions, and the white regions denote extruder start/stop points in X–Z and Y–Z external faces. Figure 6B shows a heatmap produced from micro-CT which highlights areas where red regions are the thickest extrusion and green are the thinnest. Road endpoints (white regions in Figure 6A) coincide with the locations of excess extruded material in the heatmap shown in Figure 6B.

### 3.2. Surface Morphology & Layer Cohesion

SEM evaluation revealed excellent as-sintered scaffold morphology. Inter-layer bonding appeared to be consistent, cohesive, and well-formed. Road stacking appeared to create unified walls, as shown in Figure 7, where no apparent interfacial boundary exists between roads. Layer adhesion could also be observed as a bulk quality, where the scaffolds showed sufficient durability to be handled and gripped without fracture.

Roads are smooth and surfaces are consistently corrugated (Figure 7 and Figure 8), as would be expected in material extrusion printing. Some gaps are apparent in as-sintered scaffolds (Figure 8), where a space in the as-designed surface cannot be filled with an exact number of roads at the given road diameter. In other words, the interpolated spacing cannot be filled with an integer number of roads. This is a result of the slicer calculations, not printing error.

### 3.3. Dimensional Accuracy

Figure 9 shows the dimensional accuracy of the printing and sintering process in the thirteen cubic test scaffolds compared to the as-designed side length. The printing process resulted in an increase in all dimensions and the sintering process caused a decrease (i.e., shrinkage) in all dimensions. The changes from as-designed to as-printed and from as-printed to as-sintered are expressed as percentages in Figure 10, along with the significance of these changes, where the null hypothesis assumes isotropic effects. No significant difference was observed between X and Y dimensions for printing and shrinkage, whereas Z was significantly different (*p* < 0.001) in all comparisons.

### 3.4. Scaffold Pore Size, Porosity & Wall Thickness

Micro-CT scans of three representative FKS scaffolds revealed three-dimensional characteristics relating to the pore size, porosity, and wall thickness of the finished scaffolds. The image produced by a typical scan is shown in Figure 11, and the resulting data are displayed in Table 1 below. The average as-sintered porosity of FKS scaffolds was 74.05 ± 0.38%, while the average wall thickness was 0.42 ± 0.15 mm. The pore size was evaluated as an elliptical cross section with average major and minor diameters of 2.11 ± 0.19 mm and 1.77 ± 0.14 mm, respectively.

## 4. Discussion

Scaffold breakage during the printing process posed a challenge while 3D printing FKS scaffolds. These events were the result of the nozzle colliding with previously extruded and cured roads. Therefore, a relatively high amount of operator monitoring was required throughout the course of printing FKS scaffolds, compared to gyroid scaffolds, for example. Oozing of slurry during interior road formation caused a higher (Z) road profile than expected by the program. These anomalies could be as large as a few millimeters in road length if they occurred. Consequently, if this anomaly was large enough, when the nozzle path returned to an anomalous section, it contacted a prior road and sometimes fractured the scaffold in that segment. This challenge was mitigated by very small, typically approximately 25 μm, manual adjustments to the Z-height to raise the nozzle during head movement. These adjustments were made at the operator’s discretion and varied for each scaffold due to the small deviations in extrusion discussed above. Additionally, when small collisions occurred between the nozzle and a preceding road, slurry had to be manually cleared from the nozzle tip with tweezers to prevent additional localized fractures. These Z-height adjustments and nozzle monitoring operations were typically required a half-dozen times per build and proved effective at producing scaffolds with adequate quality for BTE, with only minor imperfections. Variability will be addressed in future work by adjustments to slurry viscosity and management of residual back pressure to reduce excess slurry extrusion.

The aforementioned quality monitoring approach was also used when printing gyroid scaffolds [59], but intervention was more frequent when printing FKS scaffolds. One of the important differentiating characteristics of FKS topology compared to gyroid topology is that FKS has internal closed loop roads, which can be seen in Figure 12. Similar to excess slurry build-up on external start/stop locations at the ends of roads in each layer (described above), excess slurry also accumulated at the co-located start/stop points of circular roads in inner regions of the FKS cubes. Moreover, these loops are suspected to contribute to anisometric dimensional accuracy (shrinkage) of FKS scaffolds, as discussed below. In total, the number of starts/stops that take place are greater in printing FKS than gyroids and pose a challenge in viscous extrusion of ceramic FKS scaffolds.

A representative heatmap image from micro-CT imagery in Figure 13 shows a slice of the internal structure of a typical scaffold, where orange regions represent relatively thicker material, much as the color maps of the external faces shown in Figure 6B. The primarily green regions visible on the top of the scaffold shown in Figure 6B and throughout Figure 13 confirmed the consistent, continuous motion of extruder. However, there were small amounts of ooze in all layers, to varying degrees, usually found localized in or adjacent to the internal closed loops. In BTE applications, imperfections in the periphery of X–Z and Y–Z planes would not impede flow through the interconnected pores of the scaffolds, rendering such defects unimportant in this context. Internal pore constriction that affects BTE scaffold function will be quantified in a subsequent section.

The viscosity of the slurry is critical for build quality in viscous extrusion. The slurry needs some amount of shear-thinning behavior to extrude consistently for the entire build without clogging, but at the same time, it must be viscous enough to hold its shape and position as it bridges over the open, interconnected pores of a BTE scaffold. A less-viscous slurry exhibited greater oozing and poorer bridging than a more-viscous slurry, but a highly viscous slurry ran the risk of clogging the needle, leading to a full restart of the build. To add further complexity, slurries were observed to change viscosity over time, which we postulate is caused by a combination of the evaporation of powder-suspending agents and the exposure of resin to very small and variable amounts of light during handling. To reduce these effects, slurry was used quickly after production and exposure to ambient light was minimized. As a result, the most printable slurries favored lower viscosity at a small cost to resolution. These FKS scaffolds are sufficiently good quality for future work in BTE.

The percentage change from the as-designed to as-printed dimensions showed an increase in X, Y, and Z side lengths. This general increase likely resulted from the expansion of the slurry due to the pressure gradient from the relatively high pressures inside the nozzle chamber to the lower ambient pressures of the environment. The comparison of the X and Y expansions showed no statistically significant difference and together averaged +12.51%. We do not expect variation in X–Y because the print direction varies within each layer. The percent change in the Z direction was +6.06% and was very significantly different (*p* < 0.00001) from the X–Y data. The X–Y dimension enlarged more than Z due to ooze on the vertical (external) faces of the scaffold, as described in Figure 6. Furthermore, road flattening upon deposition, which is typical of extrusion-based 3D printing [65], reduces the Z height while simultaneously expanding in the X–Y plane, both of which contribute to these dimensional variations. The standard deviation in Z dimension change was lower than that of both X and Y, leading to the conclusion that micro-adjustments in the positive Z direction that were made by operators (discussed above) did not appear to have a significant effect on scaffold height or its variability.

The shrinkage of major dimensions between as-printed and as-sintered scaffolds occurred due to the densification of ceramic particles under heat, and it was expected from our prior work that green body dimensions would decrease by approximately 21% [59]. Again, no significant difference was seen between the effects on the X and Y dimensions, which together averaged a change of −22.74%. The Z direction showed a greater reduction at −24.73%, which was significantly different from both X and Y (*p* < 0.0001). It is postulated that the Z height shrinks the most because there is more variability in as-printed layers in Z than in X or Y due to extrusion pressure and ooze effects, as discussed above. Some dimensional variation results from the shear-thinning behavior of the slurry as it is being extruded through the nozzle, which leads to variations in flow behavior. Moreover, small amounts of material are over-extruded due to residual back pressure in the print nozzle while the print head is moving between print locations. The helical extruder that drives the viscous extrusion was backed off slightly during non-printing head movements in order to mitigate this residual backpressure, but it was not completely eliminated.

For a rectilinear scaffold, pores are consistent in shape between layers and are trivial to measure. Within a TPMS scaffold, the continuous curvature in three dimensions makes the shape and area of each pore’s cross section different on each slice as one traverses any axis. This makes pore size difficult to compare between topologies and to compare to other research. Pore size in the context of BTE seeks to quantify an interconnected channel that is conducive to the mass transport of nutrients to promote healthy bone cell activity and subsequent bone regeneration. In this context, pore size was measured as the diameter of the most-constricted inlet with the assumption that flow would occur in the Z direction. As-designed pores in the X–Y plane of FKS scaffolds are circular with a diameter of 2.42 mm. However, as-sintered pores were elliptical with average major and minor diameters of 2.11 ± 0.19 mm and 1.77 ± 0.14 mm, respectively. Shrinkage of pores occurred as a result of the sintering process and should be factored into any scaffold design, as discussed above. The percent change between as-designed to as-sintered diameters was 13.08% in the major axis and 27.10% in the minor axis. This differed from the side length shrinkage in the X–Y plane of 22.74%, which is surprising considering that material ooze would have reduced pore size further. This can be explained on the basis of the relative contributions of decreased pore size and increased wall thickness (due to settling) to the overall side length dimension. Moreover, the diameters in the major axis were aligned with the road directions, which formed the pores on each layer (Figure 14), indicating that the decrease in pore size can be attributed to layer flattening and ooze when viewed in the X–Y plane. Figure 14 illustrates the anisotropic constriction of a pore compared to the road direction overlaid on the micro-CT model of a sintered scaffold.

The as-sintered porosity of 74.05 ± 0.39% was within one standard deviation of the as-designed value of 73.71%. The porosity was open and interconnected, which is highly desirable for BTE [48,66]. This similarity results from a combination of two counter-opposing factors: the aforementioned expansion in wall thickness due to ooze, which results in decreased porosity, concurrent with the increased porosity that results from densification in sintering. The small variability in as-sintered porosity can be attributed to material ooze, which added excess material (decreasing porosity), and to minor fractures that resulted in loss of material, thereby increasing porosity. Furthermore, some porosity variation is caused by the micro-CT image processing method, which filters small variations in density differently. The average wall thickness was greater than expected and the average pore size was smaller than expected. The leads to the conclusion that as-sintered porosity would be less than as-designed porosity. The as-sintered porosity was likely inflated by the method micro-CT uses to measure it. Because a total volume cube is manually drawn in micro-CT to encompass the entire scaffold, small regions of void space are created on the perimeter of the scaffold where the greatest amounts ooze are found. This empty space would not be considered part of the enclosed void but is measured by the method, thereby increasing the measured porosity.

The as-sintered scaffolds had an average wall thickness of 0.42 mm with a standard deviation of ±0.15 mm. This wall thickness was greater than the designed value of 0.413 mm as a result of the flattening of layers and ooze that occurred during deposition, as previously described. Upon review of planar slices throughout the micro-CT images, wall thickness appeared to be bimodal, with thinner walls on the scaffold interior, and thicker walls on the exterior, agreeing with the conclusions from the color maps in Figure 6. This bimodality explains the high standard deviation of nearly ±35%, highlighting the capability of ceramic photocasting technology to produce consistent interior roads in complex topologies, which has promising implications for BTE. Future work is needed to reduce ooze at road endpoints, although it is considered less important than consistent internal morphology.

## 5. Conclusions

This study presented the development of a software algorithm for creating TPMS scaffolds suitable for 3D printing. This algorithm was then used to create scaffolds with FKS topology, which were subsequently fabricated in a ceramic commonly used in BTE. The scaffolds exhibited a porosity of 74.05%, which is considered in the ideal range of porosity for BTE [48,66]. Idiosyncrasies between printing the more common gyroid topology and the FKS topology were presented, some of which resulted in unique challenges when printing FKS scaffolds. Most of these issues were not unique to FKS but are common in any viscous extrusion 3D printing. To the best of our knowledge, this paper presents the first 3D printing of ceramic FKS scaffolds for bone tissue engineering. The software used to design cubic scaffolds in this report is available to the open-source community [62] to enable and accelerate research on TPMS, such as FKS and gyroids, and any continuous differentiable implicit function representation, also including spheres, ellipsoids, and conic sections.

The critical advantages of 3D-printed scaffolds are their tunable properties for application-specific design, their customizable shapes for patient-specific design, and their relatively rapid manufacturing rates. Results demonstrated that ceramic scaffolds with complex topologies could be 3D printed with accuracy and quality suitable for BTE. Control of porosity, pore size, and interconnectivity are particularly important, and was demonstrated in this study. The dimensional accuracy showed error, but with further research in slurry optimization and process control, reductions in ooze and improved understanding of shrinkage could enable custom ceramic FKS scaffolds for patients. And lastly, with the methods developed in this report, the entire process can occur on low-cost 3D printers in under 24 h. Future work will also include a functional grading of TPMS scaffolds along with analysis of ceramic scaffolds in the context of BTE, including the study of mechanical properties, failure modes, permeability, and in vitro osteogenic response.

## Figures and Tables

**Figure 1 jfb-14-00251-f001:**
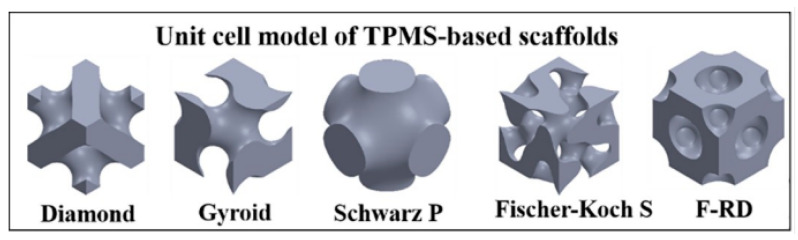
Visualization of popular TPMS scaffold unit cells [39].

**Figure 2 jfb-14-00251-f002:**
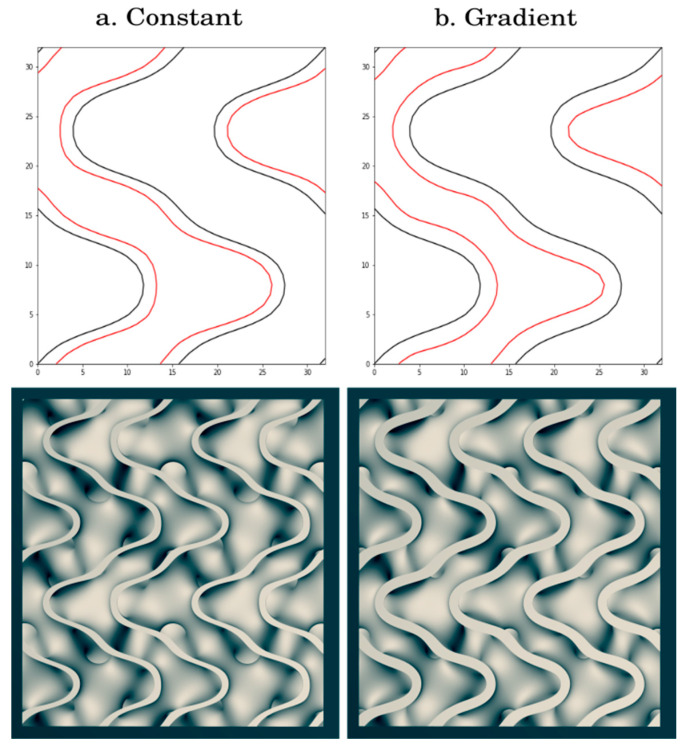
The top images show a two-dimensional slice of Fisher–Koch S at z = 0 and offsets k = 0.0 in black and k = 0.4 in red. The bottom images show a 3D rendering of FKS shelled using the respective methods. (**a**) The constant offset method with high variance in offset distance and incorrect line spacing. (**b**) Geier’s gradient offset method with uniform offset distance and incorrect line spacing.

**Figure 3 jfb-14-00251-f003:**
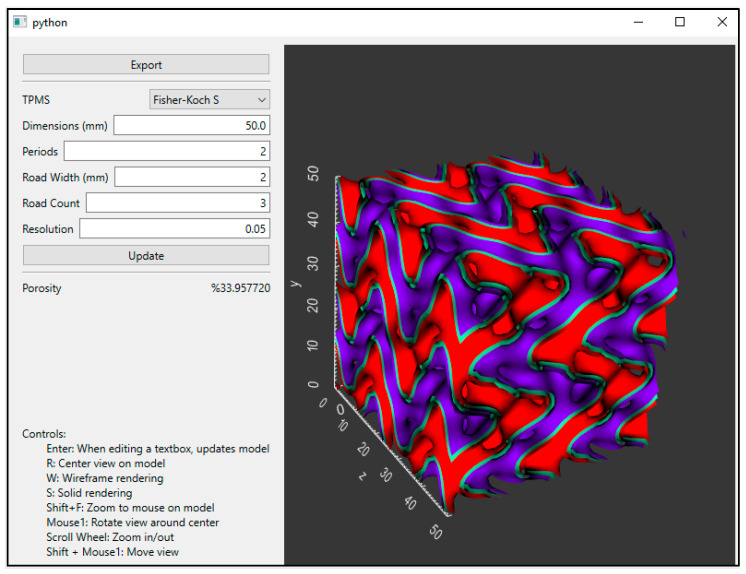
A GUI produced from the open-source code allows users to select TPMS topologies of FKS or Gyroid and adjust the properties for BTE scaffold cubes. The surface can then easily be exported as an STL file for further processing.

**Figure 4 jfb-14-00251-f004:**
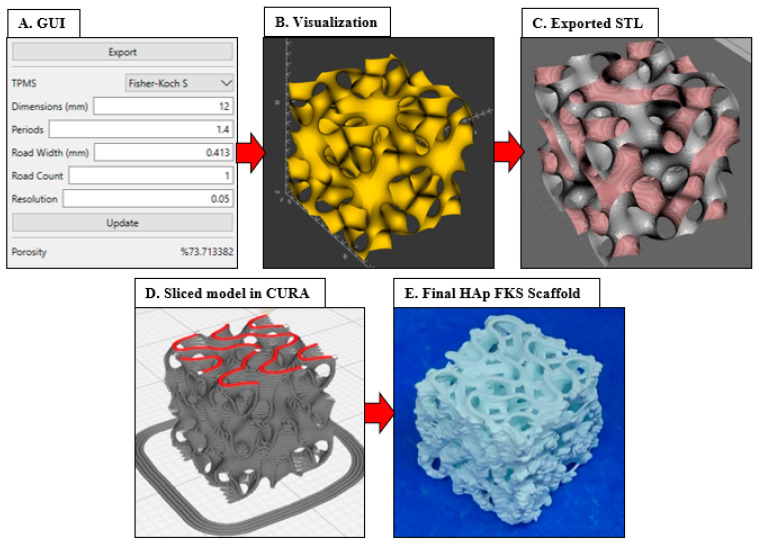
FKS scaffold fabrication workflow. The scaffold is designed for BTE (**A**,**B**), exported to an STL file (**C**), sliced in Ultimaker Cura (**D**), and photocast in Hap (**E**). Images by authors and Ultimaker Cura.

**Figure 5 jfb-14-00251-f005:**
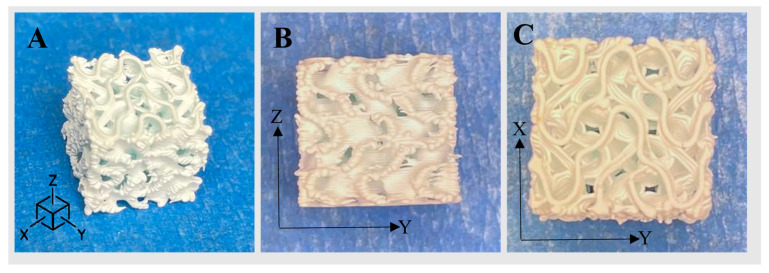
Representative images of HAp FKS scaffolds where the Z-axis is the build direction. From left to right: (**A**) isometric view, (**B**) side view, (**C**) top view.

**Figure 6 jfb-14-00251-f006:**
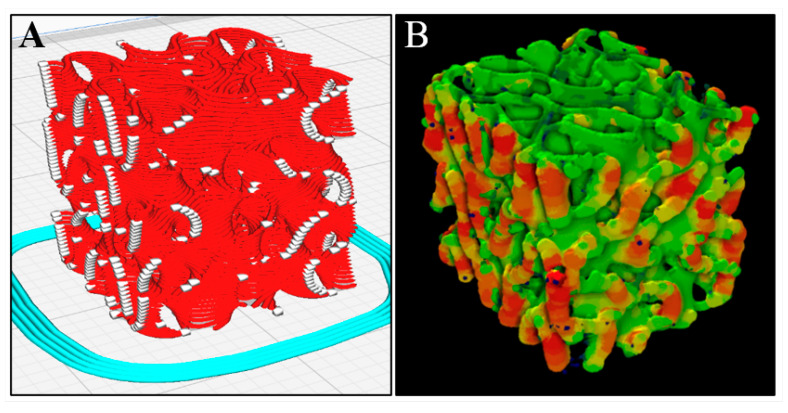
Toolpath start/stop points (white regions in (**A**)) coincide with excess extruded material shown in the micro-CT heatmap in (**B**), where red regions are thickest and green are thinnest.

**Figure 7 jfb-14-00251-f007:**
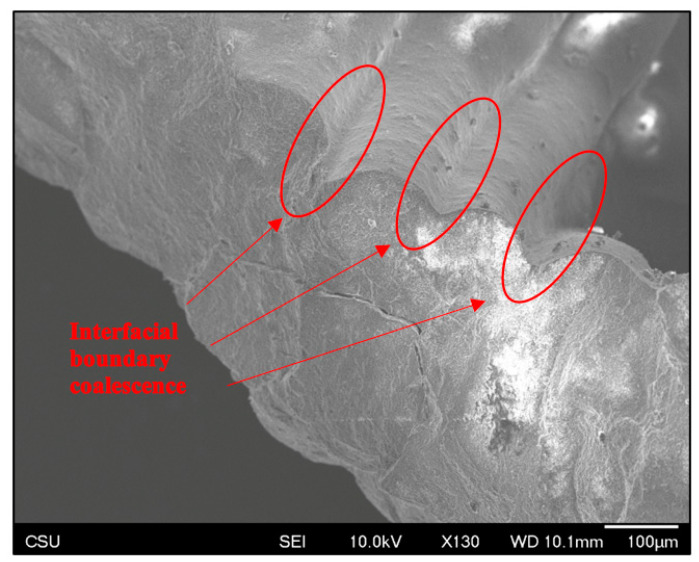
Representative SEM image of good layer cohesion and surface microstructure.

**Figure 8 jfb-14-00251-f008:**
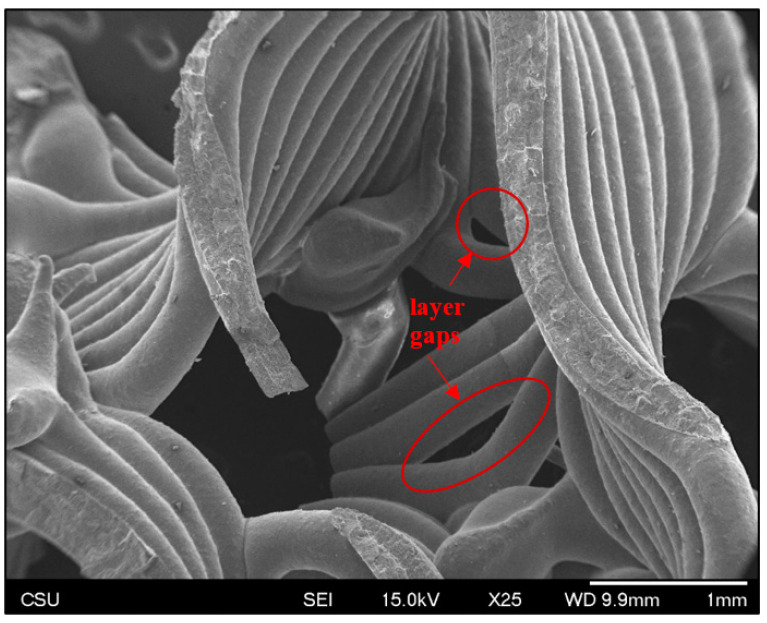
Representative SEM image showing surface microstructure and road alignment. Gaps between layers, as discussed above, can be seen in two highlighted areas.

**Figure 9 jfb-14-00251-f009:**
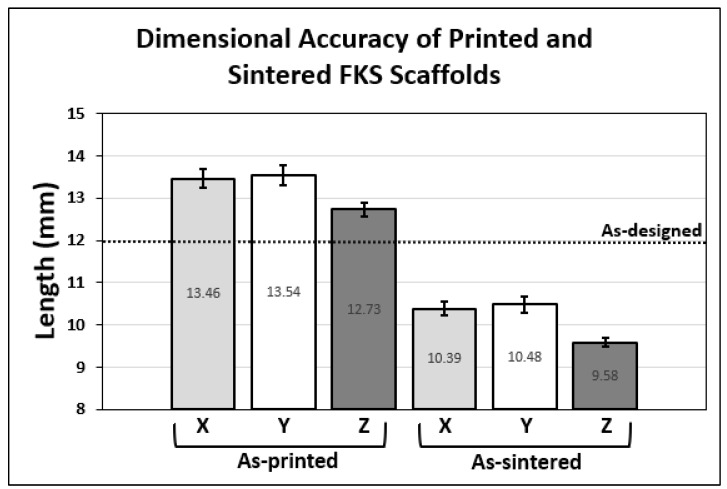
Comparing as-printed and as-sintered major scaffold dimensions with reference to the as-designed (CAD) scaffold side lengths.

**Figure 10 jfb-14-00251-f010:**
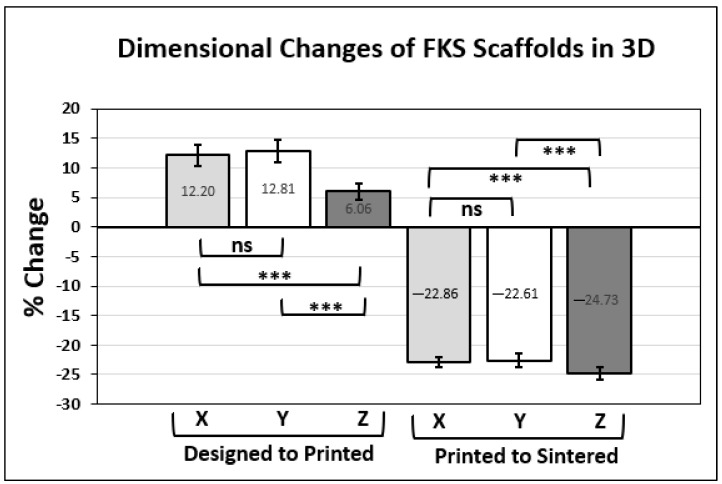
Percent change in scaffold side lengths between as-designed and as-printed is shown on the left. Percent shrinkage between as-printed and as-sintered is presented on the right. Significance comparisons (***, *p* < 0.001) are made between each major dimension, as denoted by brackets.

**Figure 11 jfb-14-00251-f011:**
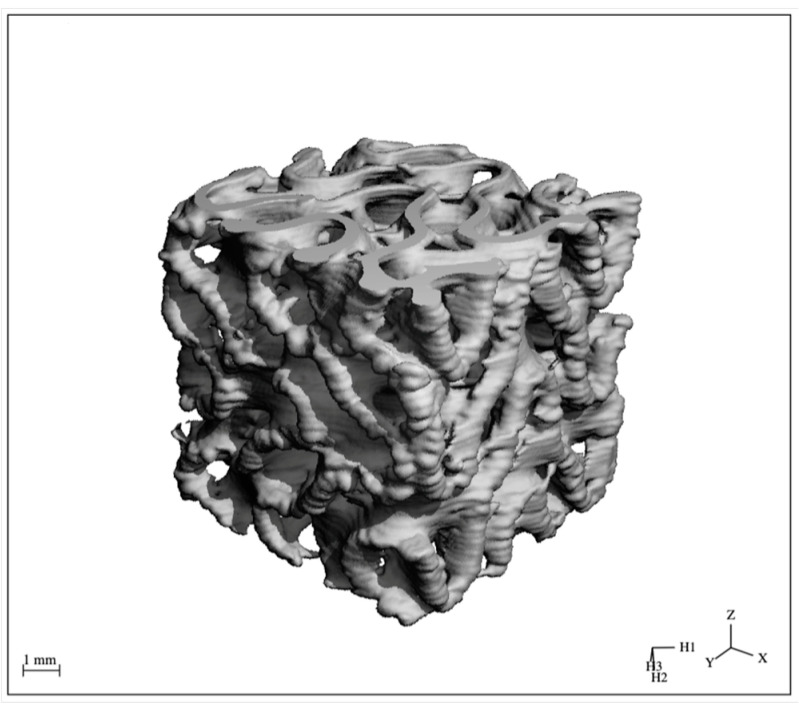
Representative μ-CT scan of a sintered HAp FKS scaffold.

**Figure 12 jfb-14-00251-f012:**
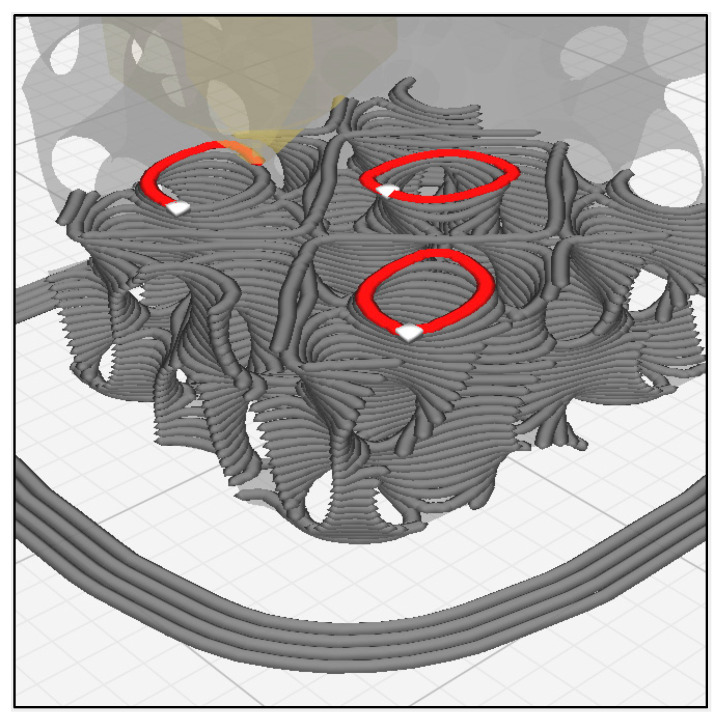
Circular extrusion paths with coincident start and stop points in the intermediate FKS layers were a common source of build errors. Image by Cura.

**Figure 13 jfb-14-00251-f013:**
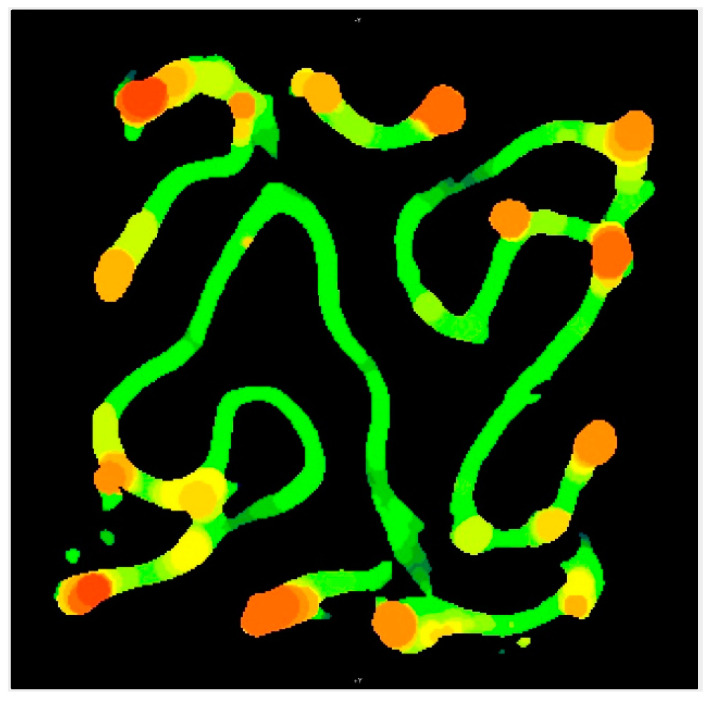
Representative micro-CT image of a cross-sectional slice in the X–Y plane of a ceramic FKS scaffold where red regions are thickest and green are thinnest.

**Figure 14 jfb-14-00251-f014:**
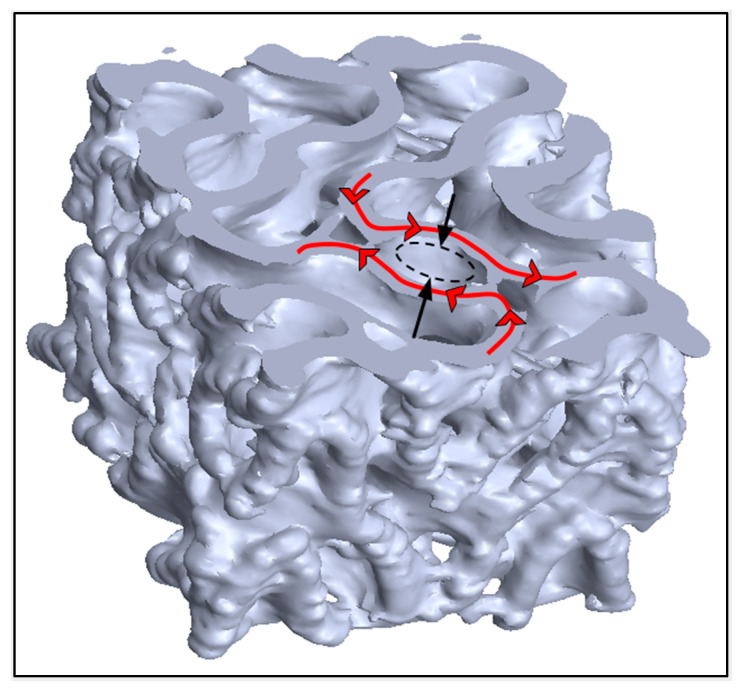
A section view in the X–Y plane of an as-sintered scaffold taken from micro-CT imagery. The construction of a pore denoted by the black arrows is perpendicular to the printed road direction denoted by the red arrows.

**Table 1 jfb-14-00251-t001:** 3D Characteristics of micro-CT imaging for FKS cubes (N = 3).

	As-Designed (Exact)	As-Sintered (Mean ± SD)	
Porosity (%)	73.713	74.05 ± 0.39	
Wall Thickness (mm)	0.413	0.42 ± 0.15	
Pore Diameter (mm) XY plane	2.423	Major	2.11 ± 0.19
		Minor	1.77 ± 0.14

## Data Availability

The data that support the findings of this study are available upon request from the authors.

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
