# Peer review of "Robocasting of Ceramic Fischer–Koch S Scaffolds for Bone Tissue Engineering"

_jfb, 2023, doi:10.3390/jfb14050251_

Round 1

Reviewer 1 Report

Dear Authors,

it has been my pleasure to read the manuscript. Unfortunately, I am not an expert in 3D printing and cannot evaluate the novelty of the design presented in the report. I mean, I have to trust the statements that the FKS morphology has not been ever probed experimentally. However, the report is otherwise very clear, convincing, and (in my opinion) does not mean to hide certain drawbacks associated with the method and its practical implementation. I enjoyed very much the practical explanations on the source of deviation of the prepared specimens from the intended mathematical model.

In fact, I have only pair of small suggestions to improve the text - otherwise my opinion that the manuscript can be accepted for publication in the present form.

1. There are too many abbreviations in the abstract. Moreover, some of them are even not explained (CAD, FEA, CFD, and 3D). I would suggest some reshaping of the abstract, probably making it a little shorter or at least avoid so many abbreviations.

By the way, in line 178 you have mentioned "FEA analyses" - but A in in FEA stands for 'analysis' itself.

2. Please check the name of the chemicals you discuss. In particular, the dash in "tri-calcium phosphate" is obsolete, and the TPO should be spelled as diphenyl(2,4,6-trimethylbenzoyl)phosphine oxide. 

As for the future research you have mentioned, I would advise to probe the morphologies with strongly different geometry parameters - for me it seems highly likely that different slurry compositions can be advantageous depending on the target structure (and probably even the macroscopic specimen size).

Good luck!

Author Response

Thank you for your time and consideration reviewing our manuscript. We have incorporated all your comments into the updated version which will be uploaded when we have completed all recommended changes. The abstract is shortened considerably. It's an improvement, so thank you.

Reviewer 2 Report

the authors adress an interesting and relevant issue in the design and morphology of bioceramic devices like HAp and TCP phosphate ceramics in bone tissue engineering. The development of software algorithms to create TPMS scaffold structures is essential, innovative and the applied methodologies correspond to the state of the art.

The results from 3D printing showed good precision and accuracy of scaffolds with complex topologies for their selected material system.

The paper is worth to be published.

The literature review seems to be not complete as recent research in design and 3 D printing of bioceramic scaffold structures from leading European research institutions (Italy, Austria, Germany etc.) are hardly mentioned. The same is true with reference to the last years important international congresses in the field.

Author Response

Thank you for your comments on our paper. We are uncertain how to proceed viz a viz your comment about our citations. A major portion of the citations we list are directly relevant, international, contemporary papers, some within 2 years. For example, I attach some of the international papers we cite that pertain specifically to TPMS/FKS scaffolds. Some are reviews. If you are aware of a paper we should have seen, please advise us. We are anxious to stay as current as possible in our work.

As you know, there are many conferences worldwide so it is impossible to track the activities of each. I searched “european congress 3D printing bioceramic scaffolds” and find no relevant papers, but please send a link to any such citations you are aware of. 

Reviewer 3 Report

This paper presents a well-composed paper addressing the development of a software algorithm for creating 600 cubic TPMS scaffolds suitable for 3D printing.

Additionally, such a topic is fascinating in the applications because in spite of huge progress in this field a great disparity remains from application specific design, to unavoidable destruction especially in rapid manufacturing rates associated with high density operation.

 Review of Literature:

The authors cited a properly composed review of the literature, and a lot of appropriate references were used in the introduction section. These statements contributed generously to the overall understanding that large bone defects are one of the biggest challenges in human orthopedic medicine leading to infections and other clinical complications, reoperations, poor functional outcomes and ultimately limb loss and the reasoning for thorough analysis and optimal design of the Triply Periodic Minimal Surfaces (TPMS) that are promising structures for bone tissue engineering.

Methodology:

The general methodology and description are well prepared. The action plan is clearly defined and well thought out except for the unpredictable time to improve the software algorithm for creating 600 cubic TPMS scaffolds. The investigation methods and techniques used to present the data for this article have been adequately selected, justified, and finally clearly discussed and concluded. However, the software algorithm were well explained, but the reliability coefficients of all presented models were not given.

Findings and conclusions

The findings were well organized, sectioned, and reported objectively. The charts were well presented, and are clear even to the average reader.

A solid point of the article is the high feasibility of using the reported public health issues’ results. The presented concept of applying the better material resulting from the presented experiment is feasible and valuable for the health care industry.

Language:

The English is almost perfect; I have not found tiny mistakes in the language.

Author Response

We are uncertain about your comment, at least to the extent that it was stated in conjunction with a comment about the software algorithm. Our algorithm is deterministic. There are few, if any, stochastic elements. While we could measure our scaffolds numerous times to measure how the dimensions vary, (i) it's not clear what measurement we would be comparing errors to (the ultimate output is some Gcode, so how do we extract a number from that) and (ii) any such variation will be very insignificant compared to the physical scale that we could possibly observe in the 3D printed scaffold. We report the dimensions of the scaffolds for the purposes of evaluating shrinkage, which is relevant to a paper about 3D printing bioceramic scaffolds, not for the purposes of evaluating the dimensional accuracy of the scaffold, which can vary by much larger scale without measurable impact on cell/bone growth outcomes.

Reviewer 4 Report

The work is well written, fits the theme of the journal and is of interest to researchers in the field.

Although it is based on the fact that HAp is frequently used in BTE and that FEA and CFD simulations have indicated that gyroid and Fischer-Koch S TPMS topologies are mechanically and structurally favorable, the study is not complete because mechanical determinations and biological tests for the material obtained by the 3D photocasting method are missing. Also, it is very likely, as the authors have shown in the discussion part and their previous works, that the parameters used are not fully optimized.

Other minor observations:

- line 285 must be EGDMA for ethylene glycol dimethacrylate;

- for multiple figures, use capital letters in the text like the notation in the figure or vice versa;

- it is not clear if "as designed" characteristics (line 278, figure 9 and related comments) already take into account the isotropic shrinkage of approximately 21% previously observed for HAp gyroids.

- the acicular shape of hydroxyapatite crystals can induce an anisotropy by an alignment of the crystals during extrusion. SEM analysis on green body can be useful.

Author Response

Please see uploaded reply. Thank you.

Reviewer 5 Report

The present work is interesting and the study is rigorous. Hence, I recommend this manuscript for publication after a minor revision as noted.

1) The authors are advised to shorten the introduction section. Moreover, please focus on the state-of-the-art as far as the experimental materials for BTE are concerned including 3D printing.

2) In the discussion section (section 4), the last paragraph should have been included in the introduction section. The authors are advised to recheck the manuscript for several repetitions of sentences/phrases.

3) Please comment on the actual use (in BTE) of the 3D-printed scaffolds as there is no study related to this.

4) Please comment on the durability of such 3D-printed scaffolds.

Author Response

Thank you for your review of our manuscript. We have addressed all of your comments in the updated version that will be uploaded this weekend. Thanks again for helping improve our work.